# Metabolomics Characterizes the Effects and Mechanisms of Quercetin in Nonalcoholic Fatty Liver Disease Development

**DOI:** 10.3390/ijms20051220

**Published:** 2019-03-11

**Authors:** Yan Xu, Jichun Han, Jinjin Dong, Xiangcheng Fan, Yuanyuan Cai, Jing Li, Tao Wang, Jia Zhou, Jing Shang

**Affiliations:** 1School of Traditional Chinese Pharmacy, China Pharmaceutical University, Nanjing 211198, China; 18260090078@163.com (Y.X.); hanjichun10@163.com (J.H.); 15077835758@163.com (J.D.); CPUFXC@163.com (X.F.); cyynora@icloud.com (Y.C.); 2School of Life Science and Technology, China Pharmaceutical University, Nanjing 210009, China; lj_cpu@126.com; 3Jiangsu Key Laboratory of Drug Screening, China Pharmaceutical University, Nanjing 211198, China; wangtao1331@cpu.edu.cn; 4Jiangsu Center for Pharmacodynamics Research and Evaluation, China Pharmaceutical University, Nanjing 210009, China; 5Qinghai Key Laboratory of Tibetan Medicine Pharmacology and Safety Evaluation, Northwest Institute of Plateau Biology, Chinese Academy of Sciences; Xining 810008, China; 6Key Laboratory of Tibetan Medicine Research, Northwest Institute of Plateau Biology, Chinese Academy of Sciences, Xining 810008, China

**Keywords:** nonalcoholic fatty liver disease, high fat-sucrose diet, metabolomics, HPLC-QTOF-MS, quercetin

## Abstract

As metabolomics is widely used in the study of disease mechanisms, an increasing number of studies have found that metabolites play an important role in the occurrence of diseases. The aim of this study is to investigate the effects and mechanisms of quercetin in high-fat-sucrose diet (HFD)-induced nonalcoholic fatty liver disease (NAFLD) development using nontargeted metabolomics. A rat model of NAFLD was established by feeding with an HFD for 30 and 50 days. The results indicated quercetin exhibited hepatoprotective activity in 30-day HFD-induced NAFLD rats by regulating fatty acid related metabolites (adrenic acid, etc.), inflammation-related metabolites (arachidonic acid, etc.), oxidative stress-related metabolites (2-hydroxybutyric acid) and other differential metabolites (citric acid, etc.). However, quercetin did not improve NAFLD in the 50-day HFD; perhaps quercetin was unable to reverse the inflammation induced by a long-term high-fat diet. These data indicate that dietary quercetin may be beneficial to NAFLD in early stages. Furthermore, combining metabolomics and experimental approaches opens avenues to study the effects and mechanisms of drugs for complex diseases.

## 1. Introduction

Nonalcoholic fatty liver disease (NAFLD) is acknowledged to be the hepatic manifestation of obesity and metabolic syndrome, including a wide spectrum of liver diseases ranging from simple fatty liver to nonalcoholic steatohepatitis (NASH), and these diseases may eventually progress to liver cirrhosis and hepatocellular carcinoma [1]. NAFLD is increasing in incidence worldwide, which is a serious threat to human health [2]. The pathogenesis of NAFLD involves many mechanisms, such as lipid peroxidation, inflammatory factor damage, oxidative stress and insulin resistance [3,4]. Lipid accumulation and inflammation have been reported to contribute to the development of NAFLD [5,6]. Several studies have discovered that some compounds, such as alpinetin and rutin, can reduce NAFLD by reducing inflammation and inhibiting lipid accumulation [7,8]. Therefore, finding a drug that can reduce inflammation and inhibit lipid accumulation may be a logical therapeutic strategy for the treatment of NAFLD.

Recent studies indicate that some small metabolites are associated with NAFLD. Research has shown that some fatty acids, such as palmitic acid [9], docosahexaenoic acid and eicosapentaenoic acid [10], changed in NAFLD and could promote lipid accumulation. Arachidonic acid could mediate inflammation involved in the development and progression of NAFLD [11]. In HFD-induced NAFLD, changed succinic acid and citric acid levels may cause disorders in the TriCarboxylic Acid cycle (TCA cycle), promote increased oxidative stress, affect liver mitochondrial function and the release of cytokines, ultimately leading to hepatitis [12]. Alterations in levels of bile acids showed a positive correlation to NAFLD in both rodents and humans [13]. Metabolomics is defined as the quantitative measurement of the time-related small metabolites of multicellular systems to pathophysiological stimuli or a genetic modification [14]. At present, metabolomics technology has been widely used in the study of NAFLD drugs such as pharmacodynamic evaluation, drug screening and mechanism research [15]. Therefore, finding and studying NAFLD drugs by using metabolomics techniques is a logical therapeutic strategy.

Quercetin is one of the most abundant dietary flavonoids, presenting various kinds of biological functions [16]. Beneficial effects of quercetin on lipid accumulation and inflammation associated with NAFLD have been reported [17]. In addition, studies have found that quercetin can effectively alleviate NAFLD injury [18]. However, the altered metabolites of quercetin on NAFLD is poorly understood. Therefore, our study aimed to explore the effects and mechanisms of quercetin on NAFLD metabolites using metabolomics technology.

## 2. Results

### 2.1. Effects of Quercetin on Liver Injury in High-Fat-Sucrose Diet (HFD)-Induced Nonalcoholic Fatty Liver Disease (NAFLD) Development

To investigate the effects of quercetin on high-fat-sucrose diet (HFD)-induced nonalcoholic fatty liver disease (NAFLD) injury, we first determined the serum aspartate and alanine transaminases (AST and ALT) levels in control, model and quercetin groups at 30 and 50 days. Liver injury was confirmed by hematoxylin–eosin (HE) staining, which demonstrated HFD-induced hepatic vacuoles, lipid droplets and hepatocyte swelling (Figure 1A) in M30 and M50 compared to the corresponding control group, and injury in M50 was more serious than in M30. Significant increases of serum AST and ALT levels (*p* < 0.01) were found in M30 and M50 compared to corresponding control group, and both levels in M50 were higher than M30 (Figure 1B,C, Appendix A). Treatment with quercetin significantly decreased AST and ALT levels in serum (*p* < 0.05) and significantly inhibited hepatic vacuoles, lipid droplets and hepatocyte swelling compared to the corresponding 30-day and 50-day model group. In addition, Q30 had better effects on HFD-induced NAFLD injury than Q50. Taken together, these data suggest that quercetin has better protective effects on hepatic injury in 30-day HFD-induced NAFLD rats than 50-day rats.

### 2.2. Metabolomic Profile of Serum Extracts in the Control, Model and Quercetin Groups from Different Time Points

To understand the reason quercetin had better protective effects on hepatic injury in 30-day HFD-induced NAFLD rats than in 50-day rats, we used High Performance Liquid Chromatography coupled to quadrupole time of flight (HPLC-QTOF)-based untargeted metabolomics. In this study, we detected 641 positive-mode features and 263 negative-mode features after peak alignment and removal of missing values. The ions with variable importance in the projection (VIP), values >1.0 and *p* < 0.05, were considered the potential differential metabolites. Two hundred and seventy-seven (277) positive-mode ions and 132 negative-mode ions significantly changed (*p* < 0.05, VIP > 1). Principal component analysis (PCA) was initially used as an unsupervised statistical method to study the metabolomic differences between control, model and quercetin groups from different time points. PCA score plots (Figure 2A,B) showed a clear separation between the control and model groups from different points, indicating that the HFD could lead to significant variations in the serum metabolic profiling, and the model group was farther from the control group at 30 days than at 50 days. In addition, the position of quercetin at 30 days was closer to the control group, indicating that quercetin could better reverse the changed metabolic patterns caused by HFD at 30 days than at 50 days (Figure 2C,D).

The differential metabolites were identified by searching the METLIN and HMDB databases and then searching the related metabolic pathway in the KEGG database. The differential metabolites contributing to the separation of the model group from the corresponding control group and reversed differential metabolites in treatment with quercetin in 30 and 50 days, along with their significance values (VIP value, and *p* value), are summarized in Appendix A.

### 2.3. Characterization of Differential Metabolite Patterns Associated with HFD-Induced NAFLD Development

We identified 19 and 17 metabolites that were significantly changed in M30 and M50 compared to the corresponding control group, respectively. Metabolites significantly changed in the model group compared to corresponding control group could be roughly divided into the following categories: (1) fatty acid related, (2) inflammation related, (3) oxidative stress related, and (4) other categories. PCA score plots analyzed in each category are shown in Figure 3A–H. There was a better separation of fatty acid-related metabolites between the control and model groups at 30 days than at 50 days, while separation of inflammation- and oxidative stress-related metabolites between the control and model groups at 50 days was better than at 30 days. PCA score plots for other metabolites showed that these metabolites could equally separate the control and model groups at both 30 days and 50 days.

As shown in the heat map (Figure 3I) and Appendix A, fatty acid-related metabolites, including adrenic acid, docosahexaenoic acid, palmitic acid, linoleic acid, oleic acid and eicosapentaenoic acid, were significantly downregulated by HFD, and these metabolites more obviously decreased in M30 than in M50. Oxidative stress- and inflammation-related metabolites, including p-cresol sulfate, indoxyl sulfate, 12(S)-HPETE and 12-HETE, were significantly upregulated, while 2-hydroxybutyric acid and arachidonic acid were significantly downregulated by HFD. P-cresol sulfate, indoxyl sulfate, 12(S)-HPETE and 12-HETE more obviously increased in M50 than in M30, while 2-hydroxybutyric acid and arachidonic acid less obviously decreased in M50 than in M30. Other increased differential metabolites in the model group compared to the corresponding control group were chenodeoxycholic acid glycine conjugate, taurocholic acid, glycocholic acid, succinic acid, 15(S)-hydroxyeicosatrienoic acid, alpha-dimorphecolic acid and 9,10,13-TriHOME. In contrast, citric acid and L-tyrosine decreased in the model group compared to the corresponding control group. Overall, the metabolites that changed in HFD-induced NAFLD at 30 days were mainly involved in fatty acids, while those that changed at 50 days were mainly involved in inflammation.

### 2.4. Metabolic Consequences of Quercetin Treatment in Rats with HFD-Induced NAFLD Development

The PCA score plots of fatty acids, inflammation, oxidative stress and other categories at 30 and 50 days are shown in Figure 4A,C. The results showed that Q30 could better reverse the changed pattern caused by the HFD whether the change was from fatty acids, inflammation, oxidative stress or other aspects. To further evaluate the reversed effect of the potential metabolites by administration of quercetin more intuitively, we detected changes in 13 potential metabolites. These data suggest that quercetin may reverse 5 fatty acids (adrenic acid, docosahexaenoic acid, palmitic acid, oleic acid and eicosapentaenoic acid), 2 inflammation-related metabolites (12(S)-HPETE and arachidonic acid), 1 oxidative stress-related metabolite (2-hydroxybutyric acid) and 5 other metabolites (15(S)-hydroxyeicosatrienoic acid, alpha-dimorphecolic acid, 9,10,13-TriHOME, citric acid and chenodeoxycholic acid glycine conjugate) in the 30-day group (Figure 4B). In contrast, quercetin could only reverse palmitic acid, 2-hydroxybutyric acid and 12(S)-HPETE in the 50-day group (Figure 4D). The effects of quercetin on differential metabolites of rats with NAFLD are shown in Appendix A. Overall, quercetin could reverse fatty acid-related metabolites in the 30-day group, while the effects on reversing inflammation related metabolites were not so obvious in the 50-day group.

### 2.5. Observation of Effects of Quercetin on Hepatic Steatosis and Inflammation in HFD-Induced NAFLD Development

To observe the effects of quercetin on lipid accumulation and inflammation in HFD-induced NAFLD at 30 and 50 days, we determined the serum lipid levels, liver lipid levels and liver inflammation in control, model and quercetin groups at 30 and 50 days. As shown in Figure 5A–D and Appendix A, compared with the corresponding control group, the levels of total triglycerides (TG), total cholesterol (TC) and low-density lipoprotein (LDL) in serum were significantly increased (*p* < 0.01), while the high-density lipoprotein (HDL) level in serum was significantly decreased in the model group (*p* < 0.01), with a higher degree in changes at 30 days compared to 50 days. Compared with the corresponding model group, treatment with quercetin decreased the levels of TG, TC and LDL in serum (*p* < 0.05) and increased the HDL level (*p* < 0.05) more at 30 days than at 50 days. In addition, liver lipid levels measured by Oil Red O-staining (Figure 5E) were consistent with serum lipid levels measured by Enzyme-linked immunoassay (ELISA). Positive Kupffer cells were observed by CD68 staining to determine the inflammation status. As shown in Figure 5F, the liver that underwent HFD showed a significant increase in the number of CD68 positive marking Kupffer cells at 50 days, while this was not so obvious at 30 days, and treatment with quercetin could not effectively inhibit this phenotype in M50. These data suggest that quercetin administration could alleviate lipid accumulation in HFD-induced NAFLD. However, it could not effectively alleviate inflammation in HFD-induced NAFLD.

## 3. Discussion

The high fat-sucrose diet model is a widely used murine model of NAFLD [19]. An HFD produces the phenotype approximating steatosis and inflammation within 4 weeks in SD rats, and these conditions continue to progress [20]. This indicates that the HFD is a good model to study NAFLD and the mechanism of action of potential therapeutic agents. In the present study, the NAFLD model in rats was successfully reproduced by feeding an HFD for 30 and 50 days, and we found that the HFD induced more serious liver injury at 50 days than at 30 days according to serum ALT, AST and HE staining. Consistent with other research results [21], we found quercetin could ameliorate liver injury symptoms caused by NAFLD through decreased serum ALT and AST and alleviation of hepatic vacuole swelling. Interestingly, Q30 had a better effect on liver injury than Q50. These results indicate that quercetin was better able to improve liver injury in NAFLD at 30 days than at 50 days.

Metabolomics, the systematic analysis of all metabolites and metabolomic pathways in a given biological system, has been increasingly recognized in the study of dietary impact [22] and drug mechanisms on metabolic diseases. As a peripheral circulation indicator that can reflect the changes in various life activities, serum has become a research topic of interest in metabolomics studies [23]. To understand the reason quercetin had better protective effects on hepatic injury in HFD-induced NAFLD rats at 30 days than 50 days, we used metabolomics technology to study this phenomenon and its potential mechanisms. The HFD led to significant variations in serum metabolic profiling in rats, which indicated metabolic disorders in the model group rats. In addition, PCA from the 30-day group showed a better separation between the control and model groups than the 50-day group. Nineteen (19) and seventeen (17) metabolites were found to significantly change between the model and control groups at 30 and 50 days, respectively, which was consistent with the PCA results. Serum metabolic profiling results showed that quercetin could better reverse the abnormal metabolic patterns caused by the HFD at 30 days than at 50 days. From the reversed metabolites, we also saw that Q30 could reverse 13 metabolites and Q50 only 3 metabolites.

The relative contents of differential metabolites associated with NAFLD and quercetin treatment were examined and then compared. The networks correlated with the potential metabolites. The main disturbed metabolic pathway related to NAFLD and the possible metabolic mechanisms of quercetin treatment are summarized in Figure 6 and Appendix A. Here, we discuss the reliable and abundant evidence for the potential differential metabolites associated with NAFLD and quercetin treatment.

This study revealed that the levels of adrenic acid, linoleic acid, docosahexaenoic acid, eicosapentaenoic acid, palmitic acid and oleic acid decreased in HFD-induced NAFLD development. The significant depletion of fatty acids indicated a reduction in fatty acid oxidation and triglyceride release from the liver, with a consequent increase in triglyceride synthesis that may contribute significantly to the development of triglyceride accumulation in hepatocytes [24]. These changes are similar to those described by metabolomic analysis in NAFLD [25]. These fatty acids decreased more significantly at 30 days than at 50 days, which indicated lipid accumulation in M30 was more serious than in M50. Quercetin could better improve NAFLD at 30 days than 50 days, perhaps due to its better effects on reversing fatty acid related metabolites, such as adrenic acid, docosahexaenoic acid, eicosapentaenoic acid, palmitic acid and oleic acid. Simultaneously, Oil Red O-staining, as well as serum TG, TC, LDL and HDL levels, confirmed the more serious lipid accumulation and better effects of quercetin at 30 days than at 50 days.

Moreover, we noticed that 12(S)-HPETE and 12(S)HETE increased while arachidonic acid decreased in HFD-induced NAFLD development. Arachidonic acid is a polyunsaturated fatty acid that can mediate inflammation and is a key player in the synthetic pathway for pro-inflammatory series 2 prostaglandins and leukotrienes [26]. 12-HPETE is produced by the nonenzymatic oxidation of arachidonic acid through the 12-lipoxygenase pathway and is metabolized to produce 12-HETE. They both participate in host defense reactions and pathophysiological conditions, such as immediate inflammation [27]. Research has shown that one biochemical pathway likely to be relevant but not yet extensively studied in NAFLD is the eicosanoid generating lipoxygenases pathway 12-LOX [28]. The lower level of arachidonic acid in the model group probably reflects body adaption to inflammation. The inflammation-related metabolites changed more significantly at 50 days than at 30 days, which indicated inflammation in M50 was more serious than in M30. Quercetin can better improve NAFLD at 30 days than at 50 days, perhaps due to its better effects on reversing inflammation-related metabolites, such as arachidonic acid and 12(S)-HPETE, while inflammation in M50 may be too serious to be reversed by Q50. Simultaneously, CD68 positive marking Kupffer cells confirmed the more serious inflammation status in NFFLD at 50 days and the better effects of quercetin at 30 days than 50 days.

We also discovered that p-Cresol sulfate and Indoxyl sulfate increased while 2-Hydroxybutyric acid decreased in HFD-induced NAFLD development. Protein-bound uremic toxins, p-cresol sulfate and indoxyl sulfate, increase oxidative stress and adversely affect chronic kidney disease progression and cardiovascular complications [29]. 2-Hydroxybutyric acid has been shown to be an early marker for both insulin resistance and impaired glucose regulation. The underlying biochemical mechanisms may involve increased lipid oxidation and oxidative stress [30]. More recently, it has been noted that elevated levels of 2-hydroxybutyric acid in the plasma are a reliable marker for early-stage type II diabetes [31]. Compared to the corresponding control group, the lower level of 2-hydroxybutyric acid in the model group represented an adaptive response to lipid oxidation or oxidative stress. The oxidative stress-related metabolites changed more significantly in 50 days than in 30 days, which indicated that oxidative stress in M50 was more serious than in M30. Quercetin could better improve NAFLD in 30 days than 50 days, perhaps due to its better effects on reversing oxidative stress-related metabolites, such as 2-hydroxybutyric acid, while oxidative stress in M50 may be too serious to be reversed by Q50.

Cholesterol is involved in primary bile acid biosynthesis to form primary conjugated bile acids such as glycocholate, taurocholate, glycochenodeoxycholate, and taurochenodeoxycholate in the liver [32]. Metabolomic analysis showed a significant increase in serum levels of glycocholate, taurocholate, and glycochenodeoxycholate in patients with NASH compared with healthy patients [33]. In HFD-induced NAFLD, the increased hepatic cholesterol may increase bile acid levels due to cholestasis, leading to an activation of lipogenic genes and the development of hepatic steatosis [34].

Alpha-dimorphecolic acid is an endogenous fatty acid peroxisomal proliferator-activated receptor-gamma agonist synthesized from linoleic acid. It can activate PPAR-gamma in human endothelial cells or adipocytes and increase plasminogen activator inhibitor type-1 expression, correlating with the risk for myocardial infarction and venous thrombosis. A study showed that alpha-dimorphecolic acid was significantly increased in NASH patients with steatosis [35]. 15(S)-Hydroxyeicosatrienoic acid (15S-HETrE) is the in vivo metabolite of gamma-linolenic acid, and it has been reported to suppress cyclooxygenase-2 overexpression and/or prostaglandin E2 biosynthesis [36] and to inhibit 3H-thymidine uptake in parallel with the upregulation of peroxisome proliferator-activated receptor-gamma expression to exert antiproliferative activities [37]. 9,10,13-TriHOME is another metabolite formed as a lipoxygenase-catalyzed product of linoleic acid. In our study, higher levels of alpha-dimorphecolic acid, 15S-HETrE and 9,10,13-TriHOME indicated lipid metabolism dysfunction in HFD-induced NAFLD.

TCA impairments are associated with multiple diseases where oxidative stress plays an important role [38]. Research showed that increased serum citric acid could increase hepatic mitochondrial biogenesis and prevent hepatic steatosis and insulin resistance [39]. Serum tyrosine was reported to be positively associated with the severity of steatosis in the liver [40] and the mechanism of tyrosine metabolism dysregulation in hepatic steatosis remains poorly elucidated. In the present study, succinic acid increased while citric acid and L-Tyrosine decreased in HFD-induced NAFLD, indicating that the TCA cycle and tyrosine metabolism were affected by HFD feeding. Quercetin could better improve NAFLD at 30 days than at 50 days, perhaps due to its better effects on reversing other metabolites, such as chenodeoxycholic acid glycine conjugate, citric acid, 15(S)-hydroxyeicosatrienoic acid, alpha-dimorphecolic acid and 9,10,13-TriHOME.

In summary, quercetin exhibited hepatoprotective activity in HFD-induced NAFLD rats at 30 days by regulating fatty acid-related metabolites (adrenic acid, docosahexaenoic acid, eicosapentaenoic acid, palmitic acid and oleic acid), inflammation-related metabolites (arachidonic acid and 12(S)-HPETE), oxidative stress-related metabolites (2-hydroxybutyric acid) and other differential metabolites (chenodeoxycholic acid glycine conjugate, citric acid, 15(S)-hydroxyeicosatrienoic acid, alpha-dimorphecolic acid and 9,10,13-TriHOME). However, quercetin could not improve NAFLD at 50 days; perhaps it could not reverse the inflammation induced by a long-term high-fat diet. Furthermore, combining metabolomics and experimental approaches open avenues to study the effects and mechanisms of drugs for complex diseases.

## 4. Materials and Methods

### 4.1. Animal Model

Experiments were performed with six-week-old male Sprague-Dawley (SD) rats weighing 180~220 g (Shanghai Super-B&K Laboratory Animal Corp. Ltd., Shanghai, China) under controlled environmental conditions (temperature of 23 ± 2 °C, relative humidity of 50−70%, and 12 h light/dark cycle) with ad libitum access to food and water. This study was approved by the Science and Technology Department of Jiangsu Province (SYXK(SU)2016-0011, 27 January 2016), and all animal experiments complied with the standard ethical guidelines under the ethical committees mentioned above. After adaptation for one week, all rats were randomly divided into six groups (*n* = 6) as follows: (a) Control group: fed with a basal diet (360 kcal/100 g; fat, 13.3 g/100 g; protein, 26.2 g/100 g; carbohydrate, 60.5 g/100 g); (b) Model group: fed with a high fat-sucrose diet (HFD) (506.8 kcal/100 g; lard, 10 g/100 g; cholesterol, 2 g/100 g; egg yolk powder, 5 g/100 g; sucrose, 10 g/100 g; propylthiouracil, 2 g/100 g; basal diet, 72.8 g/100 g); (c) Quercetin treatment group: fed with HFD and given quercetin (50 mg/kg per day). The quercetin was suspended in a 0.5% carboxymethylcellulose solution and was administered by oral gavage once per day from the twentieth day. Rats in the control group and model group were given the 0.5% carboxymethylcellulose solution only. The basal diet and high fat-sucrose diet were provided by the Jiangsu Xietong Medical and Biological Corporation (Nanjing, China). Food intake and body weight were recorded daily during the experimental period. Six rats from each group were fasted overnight, and the blood and liver were collected under anesthesia before sacrificing at 30 and 50 days.

### 4.2. Serological Analyses

AST and ALT were measured to assess the hepatic injury. TG, TC, LDL, and HDL were evaluated for lipid changes. All abovementioned indexes were measured by commercially available kits (Nanjing Jiancheng Bioengineering Institute, Nanjing, China). The experiments were performed as described previously [41].

### 4.3. Liver Histopathological Examination

HE staining was analyzed to evaluate the liver injury. Oil Red O-stained lipid droplets of the fresh liver samples were analyzed to quantify the lipid content. CD68 (1:800, ab31630, abcam, Cambridge, UK) was used for histological analysis under a light microscope Olympus-BX53. The experiments were performed as described previously [41].

### 4.4. Serum Sample Preparation

The experiments were performed as described previously [42]. Serum sample (100 μL) was mixed with 300 μL cold MeOH and then vortexed for 3 min to precipitate the protein. After centrifugation (13,300 rpm at 4 °C for 15 min), 300 μL of supernatant was transferred to another vial and then dried at room temperature using a Thermo SPD1010 SpeedVac Kit (Thermo, MA, USA). The dried residue was dissolved in 100 μL water/acetonitrile (4:1), and the solution was ultrasonicated for 15 min to dissolve the sample. After centrifugation (13,300 rpm at 4 °C for 15 min), the supernatant was transferred into the autosampler vial, and a 2 μL aliquot was injected for LC-MS analysis. For quality control, equal amounts of each sample aliquots were pooled and applied to monitor the stability of analytical performance in the platform.

### 4.5. LC-MS Analysis

The experiments were performed as described previously [42]. Serum metabolic profiling was performed using a 1260 Rapid Resolution Liquid Chromatography (RRLC) system coupled to a 6530 quadrupole time of flight (Q-TOF) MS (Agilent, Santa Clara, CA, USA). Chromatographic separations for processed serum were achieved on a XSelect HSS T3 column (2.1 × 100 mm, 2.5 μm, Waters) maintained at 40 °C and a pump flow rate of 0.4 mL/min. The mobile phase system was composed of A (water with 0.1% formic acid) and B (acetonitrile). The gradient elution program was as follows: 0–2 min, 2% B; 2–7 min, 2–60% B; 7–20 min, 60–100% B, 20–23 min, 100% B.

The electrospray ionization (ESI) source was set in positive and negative mode with the data being collected between 50–1000 *m*/*z* in centroid mode using the high-resolution mode (4 GHz). For the positive ionization mode, the MS parameters were as follows: fragmentor at 130 V, nebulization gas at 40 psi, nozzle voltage at 0 V, the vcap at 4000 V, drying gas flow rate at 8 L/min, and temperature at 325 °C, sheath gas flow rate at 12 L/min, and temperature at 350 °C. For the negative ionization mode, the parameters were the same except the nozzle voltage set to 500 V. Reference masses at *m/z* 121.050873, 922.009798 in positive mode and 112.985587, 1033.988109 in negative mode were introduced for accurate mass calibration.

### 4.6. Data Processing and Analysis

The acquired mass spectrometry data (.d) were exported to data format (.mzdata) files by MassHunter Workstation Software (Version B.06.00, Agilent Technologies). Using R Foundation for Statistical Computing, data pretreatment procedures, such as nonlinear retention time alignment, peak discrimination, filtering, alignment, and matching were performed in the XCMS package (http://metlin.scripps.edu/download/). The ion features present in less than 80% samples were screened out. Open database sources, including the KEGG, MetaboAnalyst, Human Metabolome Database, and METLIN, were used to identify metabolic pathways.

### 4.7. Statistical Analysis

Results were expressed as the mean ± SD for all pathophysiological data analyzed by Student’s *t*-test using GraphPad Prism software (San Diego, CA, USA), with *p* values less than 0.05 regarded as significant. The metabolomic variables responsible for the discrimination were identified by principal components analysis (PCA) and significant differences between the compared groups were assessed by the Mann-Whitney *U* test. A *p* value < 0.05 and VIP > 1 was considered statistically significant. PCA and heat map analysis were conducted using MetaboAnalyst (version 3.0, McGill Univeristy, Montreal, Canada, https://www.metaboanalyst.ca/).

## Figures and Tables

**Figure 1 ijms-20-01220-f001:**
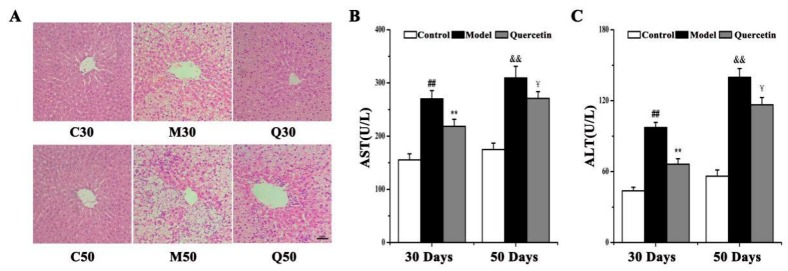
Effects of high-fat-sucrose diet (HFD) and quercetin treatment on hepatic injury in nonalcoholic fatty liver disease (NAFLD) development. (**A**) Typical hematoxylin–eosin (HE) staining (×200). White and black arrows display fat vacuole of hepatocytes and infiltration of inflammatory cells, respectively; (**B**) Serum AST and (**C**) ALT levels. Comparison was made by Student’s *t*-test, ^##^
*p* < 0.01, ^&&^
*p* < 0.01 vs. corresponding control group, ** *p* < 0.01, ^¥¥^
*p* < 0.01, ^¥^
*p* < 0.05 vs. corresponding model group. AST, aspartate aminotransferase; ALT, alanine aminotransferase.

**Figure 2 ijms-20-01220-f002:**
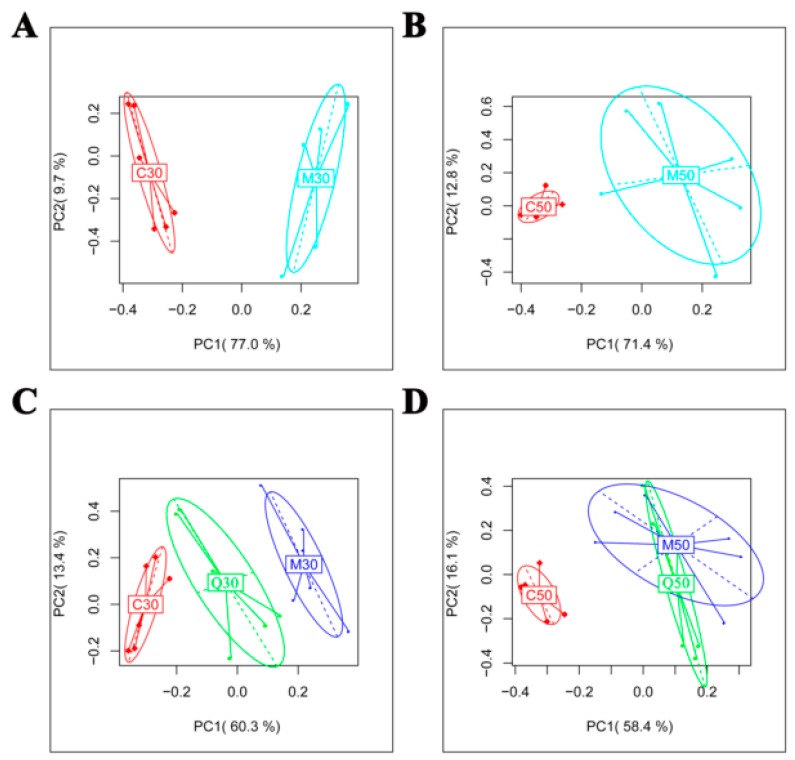
Metabolic profiling of serum samples in control, model and qudercetin groups. Principal component analysis (PCA) score plots from control and model groups after 30 days (**A**) and 50 days (**B**) feeding; PCA score plots from control, model and quercetin groups after 30 days (**C**) and 50 days (**D**) feeding.

**Figure 3 ijms-20-01220-f003:**
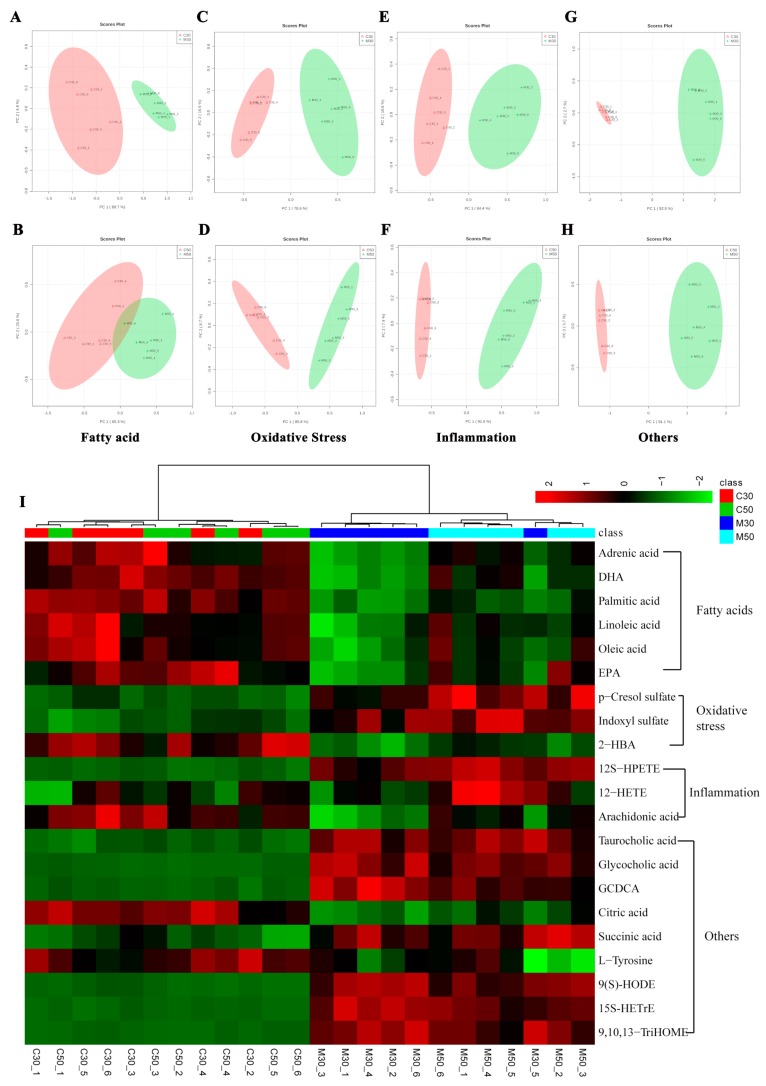
PCA score plots (**A**–**H**) and heat map (**I**) of differential metabolites in HFD-induced NAFLD development. PCA based on classification of fatty acids (**A**,**B**), oxidative stress (**C**,**D**), inflammation (**E**,**F**) and other metabolites (**G**,**H**). The colors in the heat map changing from green to red indicate more metabolites. DHA: docosahexaenoic acid; EPA: eicosapentaenoic acid; 2-HBA: 2-hydroxybutyric acid; GCDCA: chenodeoxycholic acid glycine conjugate; 9(S)-HODE: alpha-dimorphecolic acid; 15(S)-HETrE: 15(S)-hydroxyeicosatrienoic acid.

**Figure 4 ijms-20-01220-f004:**
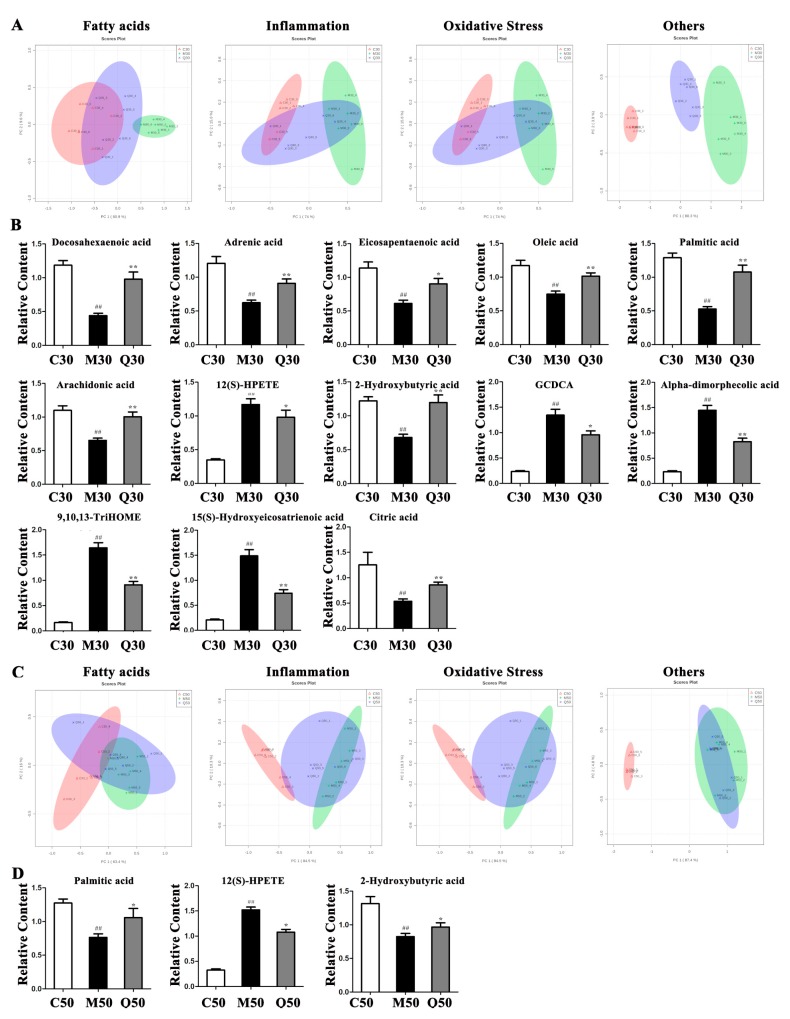
PCA score plot based on different classifications at 30 days (**A**) and 50 days (**C**). The reversed differential metabolites by quercetin at 30 days (**B**) and 50 days (**D**). ^##^
*p* < 0.01 vs. corresponding control group. * *p* < 0.05; ** *p* < 0.01 vs. corresponding model group.

**Figure 5 ijms-20-01220-f005:**
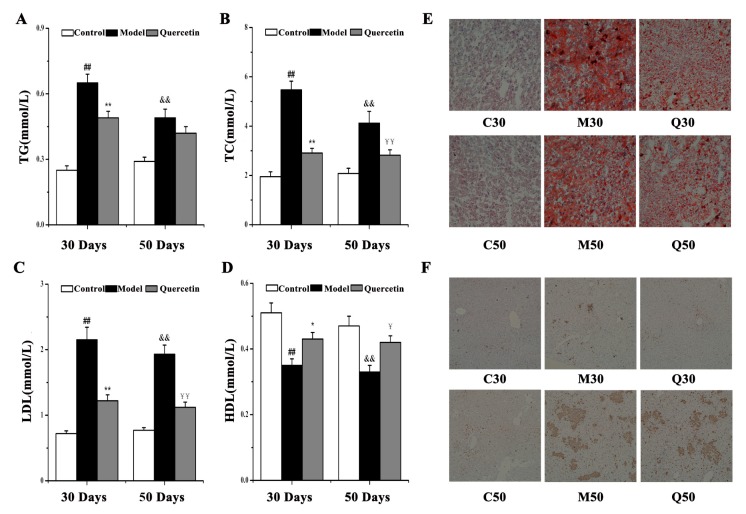
Lipid accumulation and inflammation status in control, model and quercetin groups at 30 and 50 days. (**A**) Serum TG, (**B**) TC, (**C**) LDL, (**D**) HDL levels; (**E**) Oil red O staining (×200). Small red circles indicate the formation of large cytoplasmic lipid droplets; (**F**) CD68 positive Kupffer cell (×100). Comparison was made by Student’s *t*-test, ^##^
*p* < 0.01, ^&&^
*p* < 0.01 vs. corresponding control group, ** *p* < 0.01, * *p* < 0.05, ^¥¥^
*p* < 0.01, ^¥^
*p* < 0.05 vs. corresponding model group. TG, total triglyceride; TC, total cholesterol; HDL, high-density lipoprotein; LDL, low-density lipoprotein.

**Figure 6 ijms-20-01220-f006:**
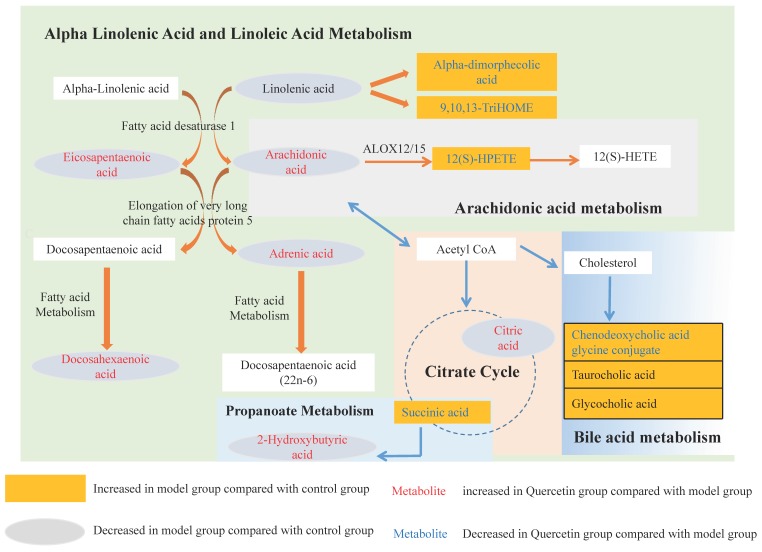
Summary scheme of quercetin-targeted pathways in rats with NAFLD at 30 days. Quercetin administration could ameliorate the fatty liver through regulating the expression of metabolites related to fatty acid and inflammation. Furthermore, the linoleic acid and arachidonic acid-related pathways could be regulated by quercetin. The red and blue metabolites represent significant (*p* < 0.05, VIP > 1) changes induced by quercetin. The red metabolites indicate quercetin-induced upregulation, and the blue metabolites indicate downregulation.

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
