# Peer review of "Metabolomics Characterizes the Effects and Mechanisms of Quercetin in Nonalcoholic Fatty Liver Disease Development"

_ijms, 2019, doi:10.3390/ijms20051220_

Reviewer 1 Report

I accept in present form.

Reviewer 2 Report

The novelty of this manuscript is clear and well documented

This manuscript is a resubmission of an earlier submission. The following is a list of the peer review reports and author responses from that submission.

Round  1

Reviewer 1 Report

Manuscript ID: ijms-429863

 Title: Metabolomics Characterizes the Effects and Mechanisms of Quercetin in the Nonalcoholic Fatty Liver Disease Development.

The novelty of this manuscript is clear and well documented. The manuscript is well written. The Tables and the Figures are clear and sufficient providing the necessary date.

Some comments are the following:

·         The authors can use the Biomarker Analysis, Pathway Analysis, Enrichment Analysis and Power Analysis on MetaboAnalyst database and get more results. It should be plentiful to add some results, such as the heatmap based on hierarchical clustering analysis, biomarker analysis results with ROC view, the correlation between quercetin exhibited hepatoprotective activity and discriminant metabolites.

·         I think it is necessary to include a table that indicates the results as the means±SEM. It not reported the definition of the indicated measurements: arithmetic means? Geometric means? Medians? [are only reported the standard deviations (SD) and probably they are arithmetic means]. The data of concentrations are normally distributed? the parametric statistical approach  is the right choice? is appropriate the comparison of arithmetical  means? We don’t know anything about that. We don’t know anything about that.

Author Response

Dear Reviewer:

  We thank you very much for giving us an opportunity to revise our manuscript, we appreciate you very much for their positive and constructive comments and suggestions on our manuscript entitled “Metabolomics Characterizes the Effects and Mechanisms of Quercetin in the Nonalcoholic Fatty Liver Disease Development” (ID: ijms-429863).

  We have studied your comments carefully and have made revision which marked in blue in the paper. We have tried our best to revise our manuscript according to the comments.
Thank you and best regards.
Yours sincerely,

Yan Xu.

The main corrections in the paper and the responds to the your comments are as following:

Answer to referee

1. The authors can use the Biomarker Analysis, Pathway Analysis, Enrichment Analysis and Power Analysis on MetaboAnalyst database and get more results. It should be plentiful to add some results, such as the heatmap based on hierarchical clustering analysis, biomarker analysis results with ROC view, the correlation between quercetin exhibited hepatoprotective activity and discriminant metabolites.

Response: Thanks for the constructive suggestions. Large-scale clinical samples are a necessary prerequisite for effective screening of biomarkers. However, animal samples were used in this experiment and the numbers are not enough big, so I didn’t make the biomarker analysis. The results based on heatmap (Figure 3I) and pathway analysis (Figure S1) have been made as following to better interpret the experiment. We have revised these in the updated manuscript and supplementary material.

Figure 3I. Heat map (I) on differential metabolites in HFD-induced NAFLD development.

Figure S1. Quercetin-targeted pathways in rats with NAFLD in 30 days.

2. I think it is necessary to include a table that indicates the results as the means±SEM. It not reported the definition of the indicated measurements: arithmetic means? Geometric means? Medians? [are only reported the standard deviations (SD) and probably they are arithmetic means]. The data of concentrations are normally distributed? the parametric statistical approach is the right choice? is appropriate the comparison of arithmetical means? We don’t know anything about that. We don’t know anything about that.

Response: Thanks for the constructive suggestions. Tables that indicate the results as the means ± SD are shown in Table S1, S4. The definition of the indicated measurements are arithmetic means. At present, the data's kurtosis is within a reasonable range (+ 1,-1). Although it is biased, its bias is within a reasonable range. According to the subsequent test, the estimation effect is acceptable. The parametric statistical approach and the comparison of arithmetical means are both appropriate. We have revised these in the updated manuscript and supplementary material.

Table S1. Effect of quercetin on serum biochemical parameters of rats with NAFLD (Mean±SD).

Parameters

TG (mmol/L)

TC (mmol/L)

LDL (mmol/L)

HDL (mmol/L)

AST (IU/L)

ALT (IU/L)

C30

0.25 ± 0.02

1.95 ± 0.2

0.72 ± 0.04

0.51 ± 0.03

155.23 ± 11.54

43.66 ± 3.06

M30

0.65 ± 0.04

5.47 ± 0.35

2.15 ± 0.19

0.35 ± 0.02

270.12 ± 15.28

97.24 ± 4.44

Q30

0.49 ± 0.03

2.91 ± 0.19

1.22 ± 0.09

0.43 ± 0.02

218.31 ± 13.45

66.29 ± 4.5

C50

0.29 ± 0.02

2.08 ± 0.21

0.77 ± 0.04

0.47 ± 0.03

174.89 ± 11.98

56.12 ± 5.24

M50

0.49 ± 0.04

4.12 ± 0.48

1.93 ± 0.14

0.33 ± 0.02

309.42 ± 22.02

139.78 ± 7.43

Q50

0.42 ± 0.03

2.82 ± 0.22

1.12 ± 0.08

0.42 ± 0.02

270.89 ± 12.94

116.56 ± 6.15

Table S4. Effect of quercetin on differential metabolites of rats with NAFLD (Mean±SD).

Metabolites

C30

M30

Q30

C50

M50

Q50

Adrenic acid

1.21 ± 0.22

0.62 ± 0.09

0.91 ± 0.14

1.30 ± 0.26

1.04 ± 0.08

0.89 ± 0.18

Docosahexaenoic acid

1.19 ± 0.2

0.44 ± 0.08

0.98 ± 0.24

1.17 ± 0.07

0.90 ± 0.13

1.02 ± 0.15

Palmitic acid

1.29 ± 0.2

0.53 ± 0.08

1.08 ± 0.23

1.27 ± 0.13

0.76 ± 0.11

1.06 ± 0.30

Linoleic acid

1.14 ± 0.2

0.78 ± 0.13

0.94 ± 0.13

1.12 ± 0.11

0.98 ± 0.09

1.04 ± 0.17

Oleic acid

1.17 ± 0.2

0.75 ± 0.10

1.01 ± 0.11

1.14 ± 0.10

0.99 ± 0.13

1.03 ± 0.17

Eicosapentaenoic acid

1.14 ± 0.2

0.61 ± 0.11

0.90 ± 0.18

1.14 ± 0.20

0.99 ± 0.16

0.98 ± 0.17

p-Cresol sulfate

0.59 ± 0.1

1.05 ± 0.21

0.81 ± 0.13

0.54 ± 0.07

1.36 ± 0.25

1.24 ± 0.24

Indoxyl sulfate

0.64 ± 0.08

1.03 ± 0.16

0.92 ± 0.17

0.62 ± 0.09

1.21 ± 0.14

1.06 ± 0.16

2-Hydroxybutyric acid

1.22 ± 0.13

0.68 ± 0.11

1.20 ± 0.25

1.32 ± 0.23

0.82 ± 0.11

0.97 ± 0.14

12(S)-HPETE

0.35 ± 0.04

1.17 ± 0.19

0.98 ± 0.24

0.32 ± 0.06

1.52 ± 0.12

1.08 ± 0.12

12-HETE

0.90 ± 0.18

0.88 ± 0.17

0.90 ± 0.17

0.83 ± 0.16

1.15 ± 0.21

1.02 ± 0.17

Arachidonic acid

1.10 ± 0.15

0.65 ± 0.08

1.10 ± 0.15

1.10 ± 0.09

0.96 ± 0.06

1.03 ± 0.12

Taurocholic acid

0.43 ± 0.08

1.25 ± 0.19

0.43 ± 0.08

0.40 ± 0.06

1.14 ± 0.16

1.09 ± 0.19

Glycocholic acid

0.15 ± 0.02

1.31 ± 0.24

0.15 ± 0.02

0.17 ± 0.03

1.08 ± 0.20

1.09 ± 0.15

Chenodeoxycholic acid glycine conjugate

0.23 ± 0.04

1.34 ± 0.26

0.23 ± 0.04

0.27 ± 0.03

0.97 ± 0.16

1.05 ± 0.13

Citric acid

1.48 ± 0.22

0.54 ± 0.10

1.48 ± 0.22

1.47 ± 0.24

0.82 ± 0.15

1.01 ± 0.14

Succinic acid

0.71 ± 0.10

1.12 ± 0.17

0.71 ± 0.10

0.61 ± 0.13

1.14 ± 0.18

1.14 ± 0.23

L-Tyrosine

1.10 ± 0.11

0.93 ± 0.13

1.06 ± 0.15

1.10 ± 0.03

0.94 ± 0.14

0.99 ± 0.19

Alpha-dimorphecolic acid

0.23 ± 0.04

1.45 ± 0.21

0.23 ± 0.04

0.17 ± 0.03

1.21 ± 0.21

1.21 ± 0.20

15(S)-Hydroxyeicosatrienoic acid

0.21 ± 0.04

1.49 ± 0.27

0.21 ± 0.04

0.19 ± 0.02

1.27 ± 0.17

1.12 ± 0.14

9,10,13-TriHOME

0.16 ± 0.03

1.64 ± 0.23

0.16 ± 0.03

0.17 ± 0.03

1.22 ± 0.21

1.25 ± 0.18

Reviewer 2 Report

Work of Yan Xu et all. "Metabolomics Characterizes the Effects and Mechanisms of Quercetin in the Nonalcoholic Fatty Liver Disease Development" is an interesting article that deals with the problem of NFLDD development. The work is clear, understandable, the goal set correctly. I recommend working for publication after adjusting and answering the following questions:

1. Has it been obtained responds to the consent of an ethics committee to work on animals?
2. Has the material - blood serum, liver been collected before or after the death of animals?
3. Why did not the authors to perform metabolic analyzes on the liver since the problem is mainly related to this organ? Do authors plan liver tests in the future?
4. How were individual metabolic features determined? Were fragmentation spectra made? I would recommend joining SM table of marked compounds with basic chromatographic and mass parameters.
5. It is worth mentioning in general the general use of metabolomics in studies of dietary impact on metabolic diseases doi: 10.1371 / journal.pone.0184798.

Author Response

Dear Reviewer:

  We thank you very much for giving us an opportunity to revise our manuscript, we appreciate you very much for their positive and constructive comments and suggestions on our manuscript entitled “Metabolomics Characterizes the Effects and Mechanisms of Quercetin in the Nonalcoholic Fatty Liver Disease Development” (ID: ijms-429863).

  We have studied your comments carefully and have made revision which marked in blue in the paper. We have tried our best to revise our manuscript according to the comments.
Thank you and best regards.
Yours sincerely,

Yan Xu.

The main corrections in the paper and the responds to the your comments are as following:

Answer to referee

1. Has it been obtained responds to the consent of an ethics committee to work on animals?

Response: Thanks for the constructive suggestions. This study was approved by the Science and Technology Department of Jiangsu Province (SYXK(SU)2016-0011) and all animal experiments complied with the standard ethical guidelines under the ethical committees mentioned above.

2. Has the material - blood serum, liver been collected before or after the death of animals?

Response: Thanks for the constructive suggestions. Six rats from each group were fasted overnight, and the blood and liver were collected under anesthesia before sacrificed. We have modified this section in our revised manuscript. (line 333).

3. Why did not the authors to perform metabolic analyzes on the liver since the problem is mainly related to this organ? Do authors plan liver tests in the future?

Response: Thanks for the constructive suggestions. Many studies suggest that metabolic analysis of serum samples can reflect metabolite levels changes in NAFLD, so we only performed the metabolic analysis of serum samples. We think your suggestion is very reasonable, and have begun to conduct metabolic analysis of liver tissue.

4. How were individual metabolic features determined? Were fragmentation spectra made? I would recommend joining SM table of marked compounds with basic chromatographic and mass parameters.

Response: Thanks for the constructive suggestions. Specific molecular masses were searched on the online databases such as Metlin (https://metlin.scripps.edu/) and HMDB (http://www.hmdb.ca/) to confirm the possible molecular composition of potential differential markers. Then MS/MS spectrometric analysis of the potential differential metabolites were operated using a QTOF mass spectrometer which provides very high mass accuracy (<10 ppm) of molecular ions and fragment ion (MS/MS) information. The structural identification results are verified by MS/MS information on the online databases such as Metlin and HMDB. The basic chromatographic and mass parameters of potential differential metabolites are shown in Table S2, S3. We have modified this section in our revised supplementary material.

 Table S2. Potential differential metabolites detected by HPLC-QTOF-MS among control, model and quercetin groups in 30 days.

Different metabolites

RT

Mass

C30 vs. M30

M30 vs. Q30

VIP

P value

VIP

P value

Adrenic acid

18.800

331.264

1.564

0.002

1.723

0.004

Docosahexaenoic acid

16.777

327.233

1.683

0.002

1.890

0.002

Linoleic acid

17.295

279.233

1.369

0.004

<< span="">1

>0.05

Eicosapentaenoic acid

15.768

301.217

1.506

0.002

1.597

0.015

Oleic acid

18.924

281.249

1.506

0.002

1.753

0.004

Palmitic acid

18.463

255.233

1.732

0.002

1.908

0.002

p-Cresol sulfate

7.159

187.007

1.435

0.002

<< span="">1

>0.05

Indoxyl sulfate

6.490

212.002

1.529

0.002

<< span="">1

>0.05

2-Hydroxybutyric acid

1.579

103.040

1.647

0.002

1.807

0.002

Arachidonic acid

17.077

303.233

1.576

0.002

1.844

0.002

12(S)-HPETE

10.713

335.223

1.686

0.002

1.0012

0.015

Chenodeoxycholic acid glycine conjugate

10.061

448.307

1.703

0.002

1.471

0.026

Taurocholic acid

8.117

514.284

1.667

0.002

<< span="">1

>0.05

Glycocholic acid

9.064

464.301

1.695

0.002

<< span="">1

>0.05

Citric acid

1.109

191.020

1.684

0.002

1.860

0.004

Succinic acid

1.495

117.019

1.470

0.004

<< span="">1

>0.05

15(S)-Hydroxyeicosatrienoic acid

13.574

321.243

1.715

0.002

1.941

0.002

Alpha-dimorphecolic acid

12.460

295.228

1.735

0.002

1.940

0.004

9,10,13-TriHOME

8.968

329.233

1.746

0.002

2.014

0.002

Statistical significance levels were determined by Mann-Whitney U test. Only metabolites with p-values<0.05 and="" vip="">1 were deemed to be statistically significant.

Table S3. Potential differential metabolites detected by HPLC-QTOF-MS among control, model and quercetin groups in 50 days.

Different metabolites

RT

Mass

C50 vs. M50

M50 vs. Q50

VIP

P value

VIP

P value

Docosahexaenoic acid

16.777

327.233

1.507

0.004

<< span="">1

>0.05

Palmitic acid

18.463

255.233

1.661

0.002

1.998

0.041

p-Cresol sulfate

7.159

187.007

1.726

0.002

<< span="">1

>0.05

Indoxyl sulfate

6.490

212.002

1.704

0.002

<< span="">1

>0.05

2-Hydroxybutyric acid

1.579

103.040

1.478

0.004

1.925

0.041

Arachidonic acid

17.077

303.233

1.265

0.009

<< span="">1

>0.05

12(S)-HPETE

10.713

335.223

1.855

0.002

3.015

0.002

12-HETE

13.108

319.228

1.202

0.026

<< span="">1

>0.05

Taurocholic acid

8.117

514.284

1.766

0.015

<< span="">1

>0.05

Glycocholic acid

9.064

464.301

1.791

0.026

<< span="">1

>0.05

Chenodeoxycholic acid glycine conjugate

9.082

448.307

1.785

0.026

<< span="">1

>0.05

Succinic acid

1.495

117.019

1.609

0.002

<< span="">1

>0.05

Citric acid

1.109

191.020

1.602

0.002

<< span="">1

>0.05

L-Tyrosine

1.478

180.067

1.128

0.026

<< span="">1

>0.05

15(S)-Hydroxyeicosatrienoic acid

13.574

321.243

1.843

0.002

<< span="">1

>0.05

Alpha-dimorphecolic acid

12.460

295.228

1.800

0.002

<< span="">1

>0.05

9,10,13-TriHOME

8.968

329.233

1.811

0.002

<< span="">1

>0.05

Statistical significance levels were determined by Mann-Whitney U test. Only metabolites with p-values<0.05 and="" vip="">1 were deemed to be statistically significant.

5. It is worth mentioning in general the general use of metabolomics in studies of dietary impact on metabolic diseases doi: 10.1371 / journal.pone.0184798.

Response: Thanks for the constructive suggestions. We have added this section in our revised manuscript (line 207).
